# Improved fuzzy logic method to distinguish between meteorological and non-meteorological echoes using C-band polarimetric radar data

Shuai Zhang[1], Xingyou Huang[1], Jinzhong Min[1], Zhigang Chu[1], Xiaoran Zhuang[1], Hengheng Zhang[2]

[1]Key Laboratory of Meteorological Disaster of Ministry of Education, Nanjing University of Information Science and Technology, Nanjing, China
[2]Institute of Meteorology and Climate Research, Karlsruhe Institute of Technology, Karlsruhe, Germany

*Correspondence to*: Xingyou Huang (huangxy@nuist.edu.cn)

**Abstract.** To obtain better performance of meteorological applications, it is necessary to distinguish radar echoes from meteorological and non-meteorological targets. After a comprehensive analysis of the computational efficiency and radar system characteristics, we propose a fuzzy logic method that is similar to the MetSignal algorithm; the performance of this method is improved significantly in weak signal regions where polarimetric variables are severely affected by noise. In addition, post-processing is adjusted to prevent anomalous propagation at far range from being misclassified as meteorological echo. Moreover, an additional fuzzy logic echo classifier is incorporated into post-processing to suppress misclassification in the melting layer. An independent test set is selected to evaluate algorithm performance, and the statistical results show an improvement in the algorithm performance, especially with respect to the classification of meteorological echoes in weak signal regions.

## 1 Introduction

Weather radar with dual-polarization capability has a wider range of application than conventional weather radar (i.e., single-polarization weather radar), in terms of providing information regarding the shape, size, spatial orientation, and physical composition of hydrometeors (Kumjian 2013a; Kumjian 2013b; Kumjian 2013c). Significant improvements have been made in meteorological and hydrological applications (e.g., data assimilation, quantitative precipitation estimation, and hydrometeor classification) after using polarimetric radar data (Giangrande and Ryzhkov 2008; Jung et al. 2008a; Jung et al. 2008b; Park et al. 2009). However, the existence of non-meteorological echoes (NMET; e.g., ground clutter (GC), anomalous propagation (AP), and clear-air echoes (CA)) in radar data often reduces the application performance. Therefore, it is necessary to separate radar data that contain meteorological echoes (MET; e.g., rain, snow, and hail) from those that contain NMET, before these applications are implemented.

Several effective algorithms for distinguishing between NMET and MET have been proposed in recent years. Lakshmanan et al. (2014) developed an algorithm based on neural networks for radar data quality control. The raw values and local

variance of polarimetric variables and Doppler moments, as well as features calculated from 3D virtual volume, are selected as neural network inputs. The output of the neural network is the MET probability at each range gate. The range gates are then clustered into contiguous regions, and the probabilities are averaged within each cluster. The average probability is compared with a preset probability threshold to determine whether the cluster is retained (considered as MET) or censored (considered as NMET). A MET–NMET classifier was developed by Tang et al. (2014) to perform reflectivity data quality

control using polarimetric radar variables and atmospheric environmental data. The algorithm combines a simple correlation coefficient filter as the primary determinant and applies a set of physically-based rules to handle some special MET (e.g., hail, non-uniform beam filling, and melting layer (ML)) and NMET (e.g., random clutter with high correlation coefficient). Krause (2016) proposed an algorithm, MetSignal, to distinguish between MET and NMET using polarimetric radar data, which has a simple design and allows users to adjust its performance according to a specific situation. The MetSignal

algorithm is based on fuzzy logic technique with a few post-processing rules; it has been selected to be implemented on the WSR-88D network in the United States. In addition, the performance of different methods in the context of distinguishing between MET and NMET is compared in Rico-Ramirez and Cluckie (2008) and Islam et al. (2012). Further, the importance of different features is also evaluated by Lakshmanan et al. (2015).

Compared with the other two methods (Tang et al. 2014; Krause 2016), the most obvious disadvantage of the neural network

method proposed by Lakshmanan et al. (2014) is the heavy computation intensity; this renders it unsuitable for operational applications, especially for radar systems with a high spatial and/or temporal resolution. Although the method proposed by Tang et al. (2014) has a higher computational efficiency, it may result in undesirable performance if applied in polarimetric radar systems with imperfect hardware technology or without noise correction (Gourley et al. 2006; Schuur et al. 2003), which is primarily attributed to excessive dependence on the correlation coefficient. Fuzzy logic is a multiple-input classifier

method that can minimize the impact from a single erroneous input. In addition, the MetSignal algorithm has the highest computational efficiency among the three methods (Krause 2016; Tang et al. 2014). Therefore, the framework of the MetSignal algorithm is adopted in this paper.

Like most methods in the context of distinguishing between MET and NMET based on polarimetric radar data, the MetSignal algorithm has high expectations for polarimetric features and sets high weights for them. However, the fluctuation

of polarimetric variables is very violent in the weak signal regions and regions affected by ML, which is not conducive to the purpose of distinguishing MET and NMET (Krause 2016; Rico-Ramirez and Cluckie 2008). The suppression of misclassification in ML regions is included in the post-processing of the MetSignal algorithm; however, the necessary consideration is lacking in weak signal regions, where polarimetric variables are severely affected by noise. The main purpose of the improved method proposed in this paper is to improve the performance of the MetSignal algorithm in weak

signal regions, hence referred as "MetSignal_noise". Additional adjustments and improvements over the MetSignal algorithm are also included in the MetSignal_noise algorithm.

The rest of this paper is organized as follows. Section 2 briefly describes the radar system used in this study and the available measurements. Subsequently, a detailed explanation of the proposed algorithm is provided in Section 3, and Section 4 presents the algorithm performance evaluation results. Finally, conclusions are provided in Section 5.

## 2 Instrument and data

The radar data used in this study were collected by a C-band dual-polarization Doppler weather radar owned by the Nanjing University of Information Science and Technology (NUIST-CDP). The main parameters of the NUIST-CDP are listed in Table 1. NUIST-CDP is designed and manufactured by Beijing Metstar Radar Company in China and is deployed at the university campus (32.21 °N, 118.72 °E). The routine scanning mode of NUIST-CDP is set to volume scanning with 14 elevation angles (0.5, 1.5, 2.4, 3.4, 4.3, 5.3, 6.2, 7.5, 8.7, 10, 12, 14, 16.7, and 19.5°) at a 7-min scan update rate. The available measurements include the reflectivity factor at horizontal polarization (Z), Doppler velocity (V), Doppler spectrum width (W), differential reflectivity (ZDR), differential propagation phase shift (PhiDP), co-polar correlation coefficient (CC), signal-to-noise ratio (SNR), and signal quality index (SQI), all at a radial range resolution of 75 m.

The NUIST-CDP data are seriously affected by GC and AP, which is attributed to the absence of clutter filtering in the signal processing. The strong CA are one of the main sources of error for some meteorological and hydrological applications (Stumpf et al. 1998; Zhang et al. 2011), which often appears in the NUIST-CDP data during the warm season. In addition, NUIST-CDP has a higher pulse repetition frequency than the operational radar (Crum et al. 1993), which implies the existence of a shorter maximum detection range and more frequent second-trip echoes. Considering that the second-trip echo is formed by meteorological targets, the algorithm temporarily classifies it as MET. The identification and removal of second-trip echoes will be considered in future research.

## 3 Method

### 3.1 MetSignal

Since a similar algorithm framework is used in both the MetSignal and the MetSignal_noise algorithms, a brief description of the MetSignal algorithm is presented first. Figure 1 summarizes the major steps of the MetSignal algorithm as a block diagram.

Except for the raw radar variables (i.e., Z, V, and CC), two texture parameters (i.e., SD(ZDR) and SD(CC)) are also input as features into the fuzzy logic echo classifier. The SD(ZDR) and SD(CC) are estimated by calculating standard deviations of ZDR and CC along a radial for 21 range gates (1.5 km in NUIST-CDP) centered on the target gate, which can characterize the magnitude of small-scale fluctuations in ZDR and CC. It is worth noting that the SD(PhiDP) input in the raw version of the MetSignal algorithm was removed to avoid texture estimation errors because of phase folding. Although there are some

conventional methods to solve phase folding (Wang and Chandrasekar 2009), they fail when applied to radar data mixed with first- and second-trip echoes.

The fuzzy logic technique is adopted in the echo classifier, which is a classification methodology widely used in the weather radar community (Gourley et al. 2007; Lin et al. 2012; Liu and Chandrasekar 2000). The additive method is applied in MetSignal algorithm to obtain the aggregation value for the MET ($A_{MET}$) to maximize the probability of detection (the multiplicative method is another way, which aims to minimize false classification; Zrnić et al. 2001):

$$A_{MET} = \frac{\sum W_x MF_x}{\sum W_x},$$                                                      (1)

where x is one of the five features mentioned above, and $W_x$ and $MF_x$ are the weights (the weight setting for each feature is shown in Table 2) and membership function value of x, respectively.

For the classification method using fuzzy logic, the membership functions selection often determines the final classification performance to a certain extent. Considering that the characteristics of radar variables depend on the specific radar systems as well as the climatological and geographical location of the radar, the membership functions are objectively determined by the statistical analysis of the NUIST-CDP measured data. Table 3 summarizes the data used for training, which are manually extracted by experienced meteorologists through a simple graphical user interface and consist of several typical events; these include GC, AP, CA, stratiform precipitation, and convective precipitation. It is worth mentioning that because there is not enough observation data as evidence, the training set does not include extremely complex situations (e.g., the boundary or transition zone between MET and NMET) to prevent the introduction of subjective bias.

The normalized frequency distributions of the features are shown in Fig. 2, which are derived using the training set. The method proposed by Cho et al. (2006) is used to determine the membership functions; it has a higher efficiency than the iterative method used in Krause (2016):

$$MF_x = \frac{FMET_x}{FMET_x + FNMET_x},$$                                                      (2)

where FMET and FNMET are the normalized frequencies of MET and NMET. The trapezoidal functions are adopted to indicate the membership functions by fitting the results of Eq. (2) using the least-squares method (the red lines in Fig. 2).

After getting the $A_{MET}$ by the calculation of Eq. (1), we compared it to a preset threshold. The target gate will be classified as MET if $A_{MET}$ exceeds the threshold; otherwise, it will be classified as NMET. Similar to membership functions, this threshold is also local, needing statistical analysis to get the optimal result. The normalized frequency distributions of $A_{MET}$, which are derived using the training set, are shown in Fig. 3a. It can be seen that there is a certain degree of overlap between the distribution of MET and NMET, and an obvious intersection is about 0.5 (the red lines in Fig. 3a). Therefore, 0.5 can be considered as an optimal $A_{MET}$ threshold of the MetSignal algorithm on the NUIST-CDP.

After obtaining the preliminary results of the fuzzy logic echo classifier, a set of post-processing rules are adopted in the MetSignal algorithm to adjust the classification results appropriately to make them more reasonable. These rules include a ZDR filter for eliminating residual CA (the range gates with absolute value of ZDR exceeding 4.5 dB are considered as

NMET), a CC filter (the range gates with CC less than 0.65 are considered as NMET), and forced classification as MET will also be performed in range gates where Z at a height of 3 km in the previous volume scan at the same location exceeds 11 dBZ, which will help to prevent misclassification in ML regions. A typical case of CA (02:23 UTC 7 May 2017) shown in Fig. 4 can well demonstrate the need for post-processing (take the ZDR filter as an example). In the field of $A_{MET}$ (Fig. 4b), many regions exceed the threshold (0.5), which will cause CA to be misclassified as MET. The primary reason for this problem is that the CA in these regions have relatively uniform ZDR and small SD(ZDR) (Figs. 4c and 4d), which causes an incorrect increase of $MF_{SD(ZDR)}$ and $A_{MET}$. Compared with the classification result directly based on the output of the fuzzy logic echo classifier (Fig. 4e), the misclassification is effectively suppressed after post-processing (Fig. 4f).

## 3.2 Improvements and adjustments in MetSignal_noise

### 3.2.1 The limitation of the use scope of V

As shown in Fig. 2b, although the V of NMET is mainly concentrated near 0 m/s while the V of MET is uniformly distributed in the whole range, there is still a large overlap between them in the regions where the absolute value of V is large. The broadening of the NMET frequency distribution is mainly attributed to the existence of CA, which is similar to that of MET in terms of V (Wilson et al. 1994). Considering that the V does not play a role in distinguishing MET from CA, some constraint conditions should be set to limit the use scope of V in the fuzzy logic echo classifier. Since the CA usually have smaller Z and larger W than GC and AP (Fang et al. 2004; Wilson et al. 1994), a 2D histogram method is adopted to analyze the NMET´s V vs. Z and V vs. W relationships in the training set to find the thresholds of Z and W for separating CA from other NMET, as far as possible. As shown in Fig. 5a, the 2D histogram of V vs. Z of NMET presents an orthogonal shape, which is composed of GC and AP with V approximately equal to 0 m/s and CA with Z below 30 dBZ and V uniformly distributed in the whole range. As shown in Fig. 5b, the 2D histogram of V vs. W of NMET is uniform overall, except for the region where V is close to 0 m/s and W is less than 2 m/s. This region is very concentrated and should be composed of GC and AP due to its static and stable characteristics. Therefore, the V is used in fuzzy logic echo classifier as a feature only when Z is greater than 30 dBZ or W is less than 2 m/s. The normalized frequency distribution and membership function of V after setting thresholds of Z and W is shown in Fig. 5c. Compared with Fig. 2b, the frequency distribution of NMET in Fig. 5c is more concentrated at 0 m/s, and the broadening has also been significantly reduced; meanwhile, that of MET remains uniformly distributed in the entire range without any notable changes.

### 3.2.2 The decrease of CC in the region of GC and AP

As illustrated in Fig. 2c, there is a significant overlap between NMET and MET in the region where CC is above 0.8, which increases the difficulty in distinguishing between NMET and MET and is also contrary to the common knowledge that NMET has a low CC (Kumjian 2013a). After analyzing a large amount of data, it is found that NMET of high CC mainly come from the GC and AP, which may be due to the characteristics of the NUIST-CDP (e.g., spatial resolution and dwell

time). Therefore, the method proposed by Zrnić et al. (2006)—CC is averaged along the radial using a 21-range gates window (1.5 km in the NUIST-CDP)—is adopted to reduce the CC of NMET with the abnormal high value. As shown in Fig. 6, it is a typical case of AP (23:53 UTC 24 May 2017) sampled by the NUIST-CDP. Compared with the raw CC in Fig. 6b, where some regions of GC and AP have high CC, the CC after average processing in Fig. 6c decreased significantly in these regions and almost all of them were below 0.9, which was expected to improve the classification performance to some extent.

The distance averaging of CC may produce some undesired side effects in the boundary region between MET and NMET, that is, the CC of MET is decreased while the CC of NMET is increased. However, their influence coverage is very limited because the window size is only 1.5 km. In addition, the impact on the averaging results will be further reduced when one of the echo types (MET or NMET) in the window accounts for a large proportion.

### 3.2.3 Improvements in weak signal regions

As shown in Fig. 7, the NUIST-CDP observed a typical case of mixed precipitation accompanied by CA within 50 km (13:24 UTC 30 May 2017). A comprehensive analysis of $A_{MET}$ (Fig. 7a) as well as Z (Fig. 7b) and SNR (Fig. 7c) reveals that MET with lower SNR near the echo edge has lower $A_{MET}$ than MET in the core regions with larger SNR, which even close to the $A_{MET}$ of CA. This is because the estimation accuracy of polarimetric variables usually depends on SNR (Bringi and Chandrasekar 2001). As shown in Figs. 7d and 7e, significant fluctuation of ZDR and decrease of CC can be observed near the echo edge. Meanwhile, their texture (i.e., SD(ZDR) in Fig. 7f and SD(CC) in Fig. 7g) has also significantly increased in these regions.

To understand the dependence between polarimetric features and SNR in more detail, a boxplot method is adopted to analyze the MET in the training set. As shown in Fig. 8a, the boxplot of SNR vs. ZDR takes the shape of a dumbbell. The broadening distribution of ZDR with the increase of SNR is attributed to the large raindrops, strong attenuation, and the resonance effect produced by hailstones, which is easy to be understood and corresponds to common knowledge (Kumjian 2013c). The MET with smaller SNR usually consists of drizzle, dry snow, and even cloud particles, which should have ZDR close to 0 dB; hence, is often used in ZDR calibration as natural targets (Ryzhkov et al. 2005). Therefore, the ZDR broadening with the decrease of SNR should not be attributed to the microphysical properties of MET, but the artifacts owing to the influence of noise. Similarly, as shown in Fig. 8b, the magnitude (dispersion) of CC decreases (increases) with decrease of SNR when SNR is less than 15 dB. These anomalies in ZDR and CC should be attributed to the weak signal affected by noise, which leads to the polarimetric variables being unable to represent the real microphysical information in MET; further, it also leads to the increase of SD(ZDR) (Fig. 8c) and SD(CC) (Fig. 8d).

Considering the dependence between polarimetric features and SNR, the polarimetric features are stratified by three SNR intervals (less than 5 dB, 5–15 dB, and larger than 15 dB), and different processing methods are used for each of these intervals. First, the data of SNR below 5 dB are directly regarded as noise and not classified. This is because the MET in this interval is extremely affected by noise and is also too weak to play an important role in meteorological and hydrological

applications. In addition, as polarimetric variables with low SNR may increase the texture of adjacent gates, the method proposed by Rico-Ramirez and Cluckie (2008)—masking the polarimetric variables of SNR below 5 dB in texture calculation—is adopted to reduce the risk of misclassification. Then, the normalized frequency distributions of polarimetric features in Fig. 2 are separated based on different SNR intervals (i.e., 5–15 dB and larger than 15 dB) and the results are shown in Fig. 9. As shown in Figs. 9a and b, the long trailing of CC of the MET caused by low SNR in Fig. 2c is well distinguished from the "normal" MET that has the CC of approximately 1 and not less than 0.8. In addition, the odd bimodal distributions of SD(ZDR) (Fig. 2d) and SD(CC) (Fig. 2e) are also well decomposed after stratification by SNR (Figs. 9c, d, e, and f), which renders the membership functions more pertinent and a better characterization of the polarimetric features is obtained at different SNR intervals.

The $A_{MET}$ obtained by the MetSignal_noise algorithm is shown as Fig. 10a. To better compare the performance of the MetSignal and MetSignal_noise algorithms, the $A_{MET}$ obtained by the MetSignal algorithm with SNR less than 5 dB is masked (Fig. 10b). The difference between them is mainly reflected in the regions of echo edge and near the radar, which is predominantly contributed by two factors. First, the fluctuation of polarimetric variables is reduced by masking the polarimetric variables of low SNR (Figs. 10c and 10d), and the texture of polarimetric variables affected by noise is significantly alleviated (Figs. 10e and 10f). Second, the polarimetric features can characterize MET and NMET more detailed by adjusting the membership functions based on different SNR intervals (Fig. 9).

Like the method used to determine the $A_{MET}$ threshold of the MetSignal algorithm, the $A_{MET}$ threshold of the MetSignal_noise algorithm is set to 0.65 based on the normalized frequency distributions shown in Fig. 3b. Compared with Fig. 3a, the distributions of MET and NMET in Fig. 3b are more focus on both ends and their overlap is significantly reduced, which can also show that the MetSignal_noise algorithm has a better classification performance than the MetSignal algorithm.

### 3.2.4 Post-processing adjustments for ML regions

The last step in the post-processing of the MetSignal algorithm is to check the constant-altitude plan position indicator (CAPPI) of Z at 3 km in the previous volume scan. The range gates will be force-classified as MET if the CAPPI at the same location exceeds 11 dBZ; this aims to prevent the misclassification in ML regions. However, due to the strong super-refraction caused by specific weather conditions (Doviak and Zrnić 2006), the NUIST-CDP sometimes detects AP more than 11 dBZ at a far range (corresponding to an altitude higher than 3 km), which will misclassify AP as MET after post-processing. Fig. 11 shows the same AP case as in Fig. 6. Although the $A_{MET}$ obtained by the MetSignal algorithm (Fig. 11a) has a low value (i.e., the classification result is more likely to be NMET), there are still many range gates misclassified as MET in the final result (Fig. 11b) due to improper post-processing. In consideration of the potential risk of misclassifying AP into MET in this post-processing step, this post-processing rule has been removed in the MetSignal_noise algorithm.

However, the lack of special precaution in the ML regions causes a frequent misclassification occurrence, because MET in ML regions and NMET have similar characteristics in polarimetric features. Therefore, an additional fuzzy logic echo

classifier, without the polarimetric features input, is implemented in the post-processing of the MetSignal_noise algorithm in the potential ML regions (initially defined as the regions over 2.5 km in height based on the statistical analysis using the training set), for range gates classified as NMET in the first fuzzy logic echo classifier. Considering that AP in the potential ML regions may not be classified effectively by solely using Z and V, the SD(Z) (using the same estimation method as SD(ZDR) and SD(CC)) is input into the additional fuzzy logic echo classifier, to improve the classification performance.

Figure 2f shows the normalized frequency distribution and membership function of SD(Z), which are also derived from the training set, but only use data in the potential ML regions. If these range gates (classified as NMET in the first fuzzy logic echo classifier in the potential ML regions) are classified as MET in the additional fuzzy logic echo classifier, then these are highly probable to be influenced by ML and should be reclassified as MET.

As shown in Fig. 12, the NUIST-CDP observed a typical case of stratiform precipitation (10:49 UTC 23 May 2017). Although the bright band characteristic of Z is not very obvious (Fig. 12d), the location of the ML region at a range of about 100 km can be well revealed by ZDR and CC (Figs. 12e and 12f). As shown in Fig. 12b, the $A_{MET}$ before post-processing (the result of the first fuzzy logic echo classifier) has an obvious low value in the ML region, due to the similar characteristics of polarimetric features between MET in ML regions and NMET. The $A_{MET}$ after post-processing (the range gates classified as NMET in the first fuzzy logic echo classifier will be substituted for the $A_{MET}$ of the additional fuzzy logic echo classifier) is shown in Fig. 12a; the abnormal decrease of $A_{MET}$ before the post-processing in the ML region is effectively suppressed. The final classification result (Fig. 12c), based on $A_{MET}$ after post-processing, shows good performance in the ML region.

In addition to the ML region, some other special MET with abnormal polarimetric features will also cause the misclassification of the algorithm (e.g., the threshold of the ZDR filter setting in the post-processing could be reached if wet hailstones are inside the radar sample volume). Therefore, the additional fuzzy logic echo classifier can also mitigate these problems to some extent by eliminating polarimetric features in the input.

## 4 Evaluation

To objectively evaluate the MetSignal_noise algorithm performance and its improvement compared with the MetSignal algorithm, a test set independent of the training set is selected and listed in Table 4 (the same extraction method as the training set). Two methods were used to compute skill: the fraction of correct classification for each echo type (FCC) and the overall Heidke skill score (HSS; Doswell et al. 1990), which is computed as:

$$HSS = \frac{2(ad-bc)}{(a+c)(c+d)+(a+b)(b+d)},$$ (3)

where $a$ represents the number of hits, $b$ the false alarms, $c$ the misses, and $d$ the correct nulls.

The skill results of the MetSignal and MetSignal_noise algorithms based on the test set are shown in Table 5. To facilitate the analysis of the dependence between the classification performance and SNR, the computation of the classification skill is

stratified by three SNR intervals (larger than 15 dB, 5–15 dB, and larger than 5 dB). By comparing the classification performance of the two algorithms in MET, it is found that the classification skill of the MetSignal_noise algorithm is significantly higher than that of the MetSignal algorithm, especially in the SNR interval greater than 5 dB and less than 15 dB. This can verify that the MetSignal_noise algorithm can improve the classification performance of the MetSignal

algorithm at low SNR by stratifying polarimetric features based on SNR intervals, and masking low SNR polarimetric variables in texture calculation. The better performance of the MetSignal_noise algorithm in the SNR interval greater than 15 dB is mainly owed to the fact that membership functions of polarimetric features are more targeted after SNR stratification. Compared with the difference of classification skill in MET between the two algorithms, the difference in NMET is smaller. The classification performance of the MetSignal algorithm in NMET is slightly better than that of the MetSignal_noise

algorithm in the SNR interval greater than 5 dB and less than 15 dB, which should be attributed to the misclassification of CA into MET in the potential ML regions after post-processing because non-polarimetric features (i.e., Z, V, and SD(Z)) cannot effectively distinguish CA from MET (Lakshmanan et al. 2007; Tang et al. 2014). The main reason for the lower classification skill of the MetSignal algorithm in the SNR interval greater than 15 dB is that the Z of AP is sometimes more than 11 dBZ at 3 km, and will be classified as MET in the post-processing of the MetSignal algorithm.

In addition to evaluating the performance of the MetSignal and MetSignal_noise algorithms, it is also necessary to perform a sensitivity analysis of the four improvement steps in the MetSignal_noise algorithm, i.e., ignore one of the improvement steps and analyze its impact on classification performance. All sensitivity analyses were performed in the range of SNR larger than 5 dB, and included four parts: 1) SA1 (without the limitation of the use scope of V); 2) SA2 (without the decrease of CC in the region of GC and AP); 3) SA3 (without the improvements in weak signal region); and SA4 (without the

adjustments of post-processing for ML region). The results of the sensitivity analysis are summarized in Table 5. Compared with SA3 and SA4, the SA1 and SA2 have less influence on the skill scores of the MetSignal_noise algorithm. The main reason is that the first two improvement steps mainly focus on NMET, but the classification performance of the MetSignal algorithm on NMET is very satisfactory (the difference of $FCC_{NMET}$ between MetSignal algorithm and the MetSignal_noise algorithm is only 0.9 % in the range of SNR larger than 5 dB). In addition, the weights of V and CC in the fuzzy logic echo

classifier are small. Although the MetSignal_noise algorithm focuses on the improvement in the weak signal region, SA3 does not have a substantial decrease in the skill scores compared to the MetSignal algorithm. The main reason is that the additional fuzzy logic echo classifier in the post-processing can correctly reclassify the misclassified MET in potential ML regions, where weak signal echoes appear frequently. The decrease of $FCC_{MET}$ in SA4 is mainly attributed to the misclassification in ML regions. However, due to the limited data affected by ML, the magnitude of the decrease is not very

notable. The reason for the decrease of $FCC_{NMET}$ in SA4 is the same as the MetSignal algorithm; that is, some of AP is misclassified as MET after post-processing (MetSignal algorithm). From the HSS of the sensitivity analysis, it can be seen that ignoring any improvement step will reduce the overall score. Therefore, all the improvements have a positive effect on the classification performance, even if some improvements do not play an important role.

**5 Conclusions**

An improved fuzzy logic method, MetSignal_noise, is proposed in this paper to distinguish between MET and NMET using polarimetric radar data from the NUIST-CDP. The most significant improvement over the raw version (MetSignal) is its better classification performance in weak signal regions by stratifying polarimetric features based on SNR intervals and masking low SNR polarimetric variables in texture calculation. In addition, the thresholds of Z and W are set to limit the scope of V in order to improve its classification performance and prevent its contribution to the misclassification of CA. An

averaging method along the radial is also used to decrease the abnormally high value of CC from GC and AP. The post-processing rule used to prevent misclassification in ML regions in the MetSignal algorithm sometimes reclassifies AP at far range into MET; therefore, it has been removed in the MetSignal_noise algorithm, and substituted by an additional fuzzy logic echo classifier without the polarimetric features input.

    An independent test set is selected to evaluate the algorithm performance; the results show that the MetSignal_noise

algorithm is overall better than the MetSignal algorithm, especially in low SNR regions. However, the MetSignal_noise algorithm is slightly worse than the MetSignal algorithm in SNR intervals greater than 5 dB and less than 15 dB. This is because some CA are reclassified as MET after post-processing because non-polarimetric features cannot effectively distinguish CA from MET. Although increasing the height threshold of the potential ML region can improve this defect as CA does not usually appear at high altitudes (Wilson et al. 1994), this will cause some low ML regions to miss the post-

processing. At present, a CA identification method based on radial continuity is under development, which is expected to greatly reduce the risk of CA misclassification. The altitude of ML depends on the season and geographical location (Zhang and Qi 2010). Therefore, real-time ML identification algorithms (Giangrande et al. 2008; Zhang et al. 2008) or atmospheric environmental data (Tang et al. 2014) have been considered as additions to the MetSignal_noise algorithm in the following study to select a better height threshold. Moreover, the advanced clutter suppression algorithm based on signal processing

(Hubbert et al. 2009a; Hubbert et al. 2009b; Torres et al. 2014) should be considered to be introduced into the NUIST-CDP. This is because when the ML appears at a lower altitude (frequently occurs in winter precipitation events), it will be fully mixed with the ground clutter. Then even if the height of ML is accurately located, the ML region may still be misclassified as NMET. The sensitivity analysis of the MetSignal_noise algorithm shows that all the improvements have a positive effect on the classification performance, even if some improvements are not significant. In addition, we plan to identify and

eliminate the second-trip echo in a future work to further improve data quality.

*Acknowledgements.* This work was supported by the National Key Technologies R & D Program of China through Grant 2017YFC1502103 and the National Natural Science Foundation of China through Grant 41430427. We acknowledge the investigators and operators of NUIST-CDP. We also want to thank Chian Zhang of Beijing Metstar Radar Company in

China for providing constructive comments on this manuscript.

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

**TABLE 1: Main parameters of NUIST-CDP.**

| Parameters | NUIST-CDP |
| --- | --- |
| Transmitter | Klystron (5,600 MHz) |
| Pulse width | 0.5 us |
| PRF | 1,000 Hz |
| Peak power | 250 kW |
| Receiver | Simultaneous Horizontal/Vertical |
| Noise figure | 3 dB |
| Dynamic range | 90 dB |
| Sensitivity | -109 dBm |
| Antenna feeder | Paraboloid |
| Antenna gain | 48.5 dB |
| Reflector diameter | 8.5 m |
| Beam width | 0.54° |


**TABLE 2: The weight setting for each feature in the MetSignal algorithm.**

| Feature | Weight |
|---------|--------|
| Z | 1 |
| V | 1 |
| CC | 1 |
| SD(ZDR) | 2 |
| SD(CC) | 1 |


**TABLE 3: List of events used for training membership functions (UTC).**

| Date | Description |
| --- | --- |
| 00:00–01:00 1 May 2017 | AP |
| 01:00–02:00 2 May 2017 | stratiform precipitation |
| 04:00–05:00 3 May 2017 | CA |
| 14:00–15:00 3 May 2017 | convective precipitation |
| 18:00–19:00 4 May 2017 | GC |
| 20:00–21:00 5 May 2017 | stratiform precipitation |
| 00:00–01:00 6 May 2017 | GC |
| 08:00–09:00 6 May 2017 | stratiform precipitation |
| 06:00–07:00 7 May 2017 | CA |
| 18:00–19:00 7 May 2017 | convective precipitation |
| 22:00–23:00 8 May 2017 | GC |
| 01:00–02:00 11 May 2017 | AP |
| 09:00–10:00 11 May 2017 | convective precipitation |

| | |
|---|---|
| 17:00–18:00 11 May 2017 | stratiform precipitation |
| 13:00–14:00 12 May 2017 | CA |
| 22:00–23:00 13 May 2017 | AP |
| 06:00–07:00 14 May 2017 | convective precipitation |
| 17:00–18:00 14 May 2017 | stratiform precipitation |
| 01:00–02:00 15 May 2017 | CA |
| 22:00–23:00 16 May 2017 | AP |
| 14:00–15:00 18 May 2017 | CA |
| 09:00–10:00 19 May 2017 | stratiform precipitation |
| 08:00–09:00 20 May 2017 | convective precipitation |

**TABLE 4: List of events used for evaluating algorithm performance (UTC).**

| Date | Description |
|------|-------------|
| 04:00–05:00 21 May 2017 | CA |
| 09:00–10:00 22 May 2017 | convective precipitation |
| 05:00–06:00 23 May 2017 | stratiform precipitation |
| 22:00–23:00 23 May 2017 | GC |
| 22:00–23:00 24 May 2017 | AP |
| 22:00–23:00 26 May 2017 | AP |
| 12:00–13:00 28 May 2017 | CA |
| 05:00–06:00 30 May 2017 | stratiform precipitation |
| 10:00–11:00 30 May 2017 | convective precipitation |
| 22:00–23:00 31 May 2017 | stratiform precipitation |

**TABLE 5: The classification performance of MetSignal and MetSignal_noise algorithms, and the sensitivity analysis of different**

**improvement steps in MetSignal_noise algorithms. All skill scores were computed based on data in test set.**

| | MetSignal | | | MetSignal_noise | | | sensitivity analysis | | | |
|---|---|---|---|---|---|---|---|---|---|---|
| | 5 < SNR < 15 [dB] | SNR > 15 [dB] | SNR > 5 [dB] | 5 < SNR < 15 [dB] | SNR > 15 [dB] | SNR > 5 [dB] | SA1 | SA2 | SA3 | SA4 |
| $FCC_{MET}$ | 79.8 % | 98.7 % | 86.8 % | 99.2 % | 99.7 % | 99.4 % | 99.1 % | 99.1 % | 95 % | 97.1 % |
| $FCC_{NMET}$ | 95.8 % | 96.4 % | 96.2 % | 94.9 % | 98.4 % | 97.1 % | 96.6 % | 96.8 % | 96.6 % | 96.4 % |
| HSS | 0.756 | 0.95 | 0.83 | 0.94 | 0.981 | 0.965 | 0.957 | 0.959 | 0.916 | 0.935 |

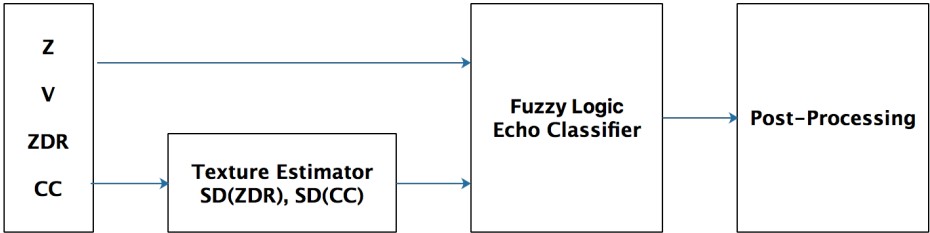

**Figure 1: Block diagram of the MetSignal algorithm.**

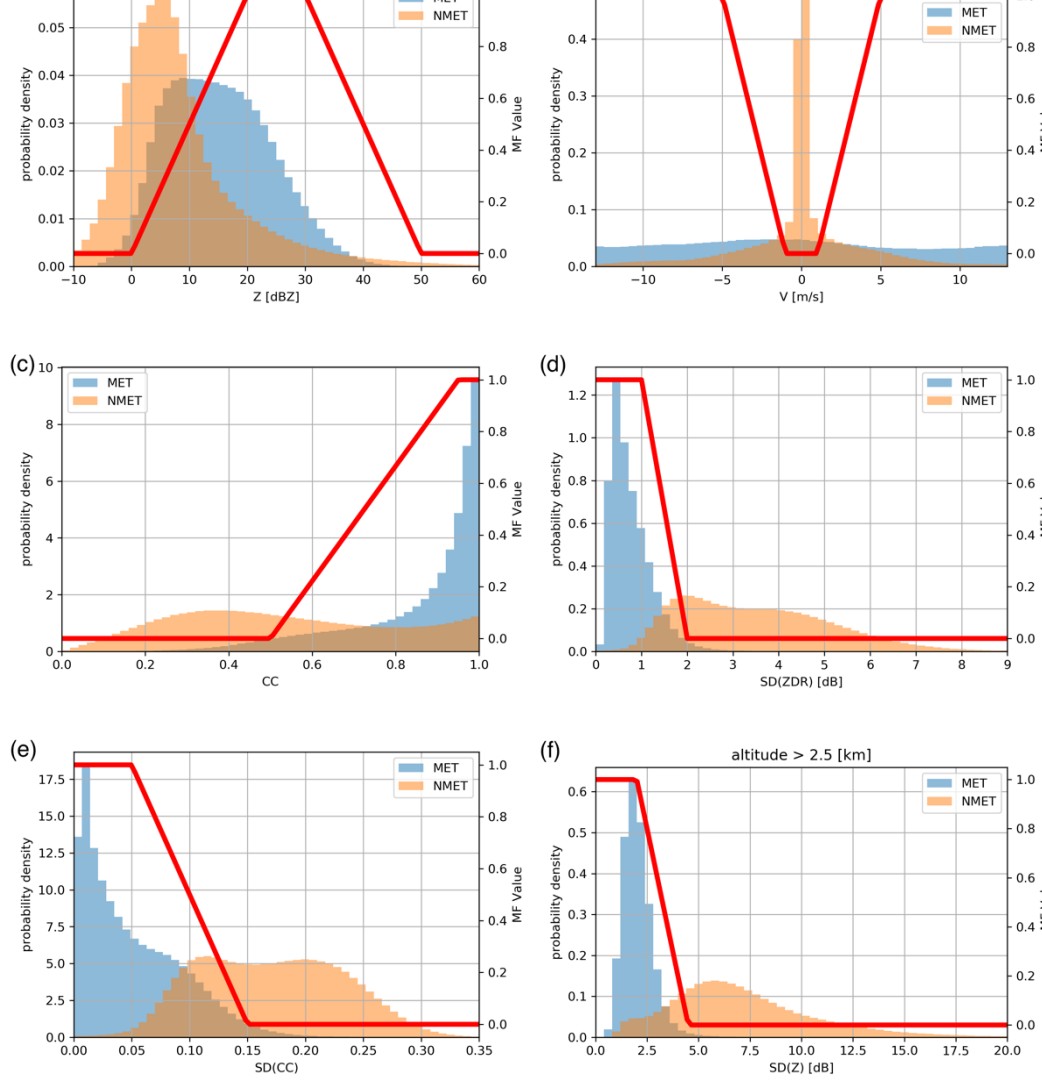

**Figure 2: The normalized frequency distributions and membership functions of features. (a) Z, (b) V, (c) CC, (d) SD(ZDR), (e) SD(CC), and (f) SD(Z) (in the regions over 2.5 km in height).**

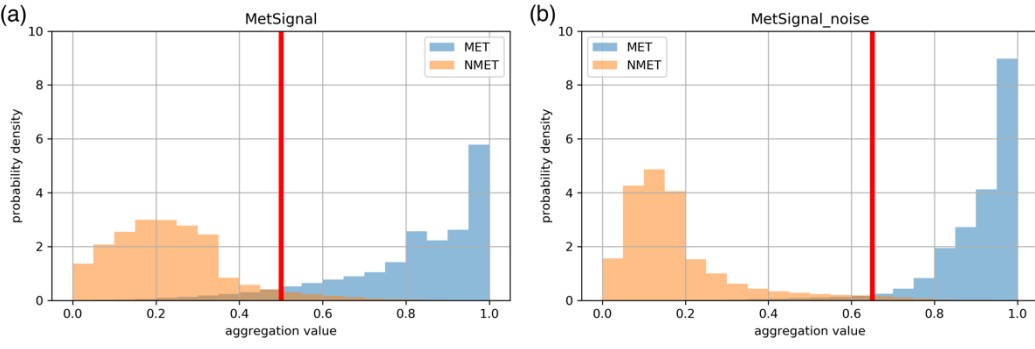


**Figure 3: The normalized frequency distributions and thresholds of A$_{\text{MET}}$. (a) MetSignal, and (b) MetSignal_noise.**

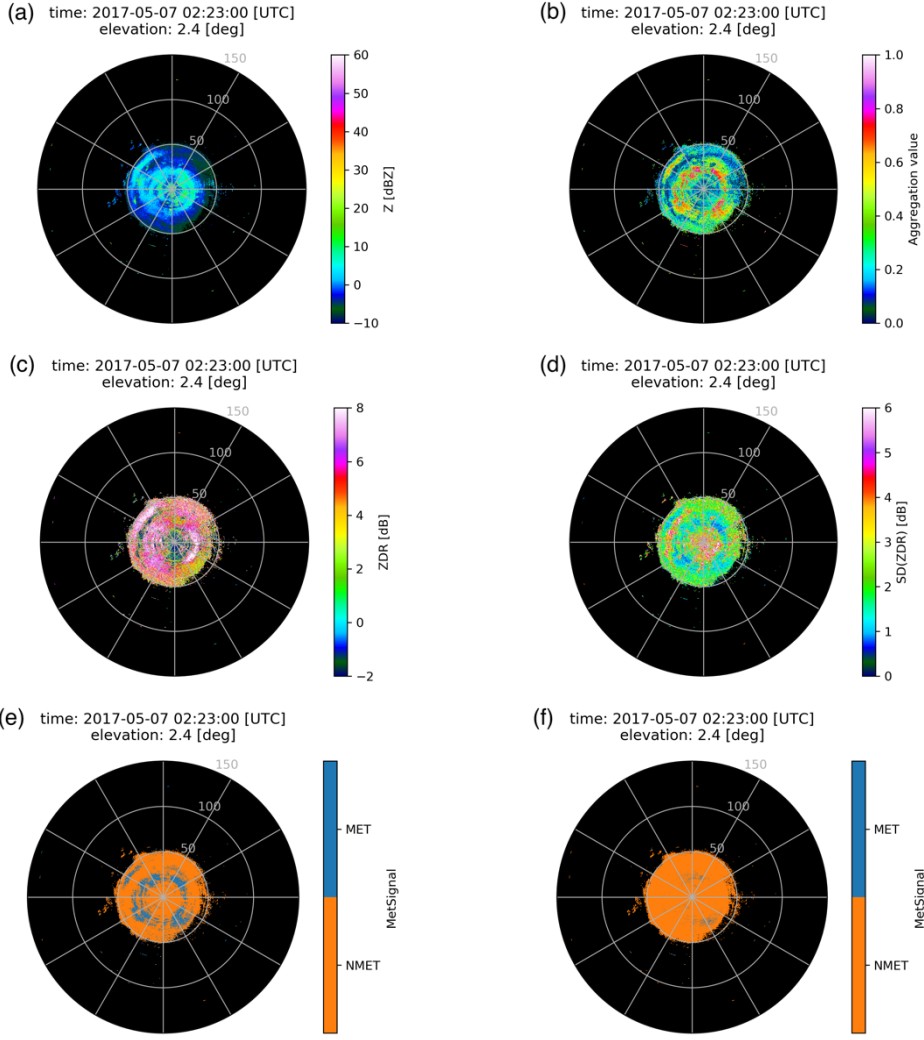


**Figure 4: (a) Z, (b) A$_{MET}$ (MetSignal), (c) ZDR, (d) SD(ZDR), (e) MetSignal before post-processing, and (f) MetSignal after post-processing. All from the NUIST-CDP at 02:23 UTC 7 May 2017 from an elevation of 2.4°.**

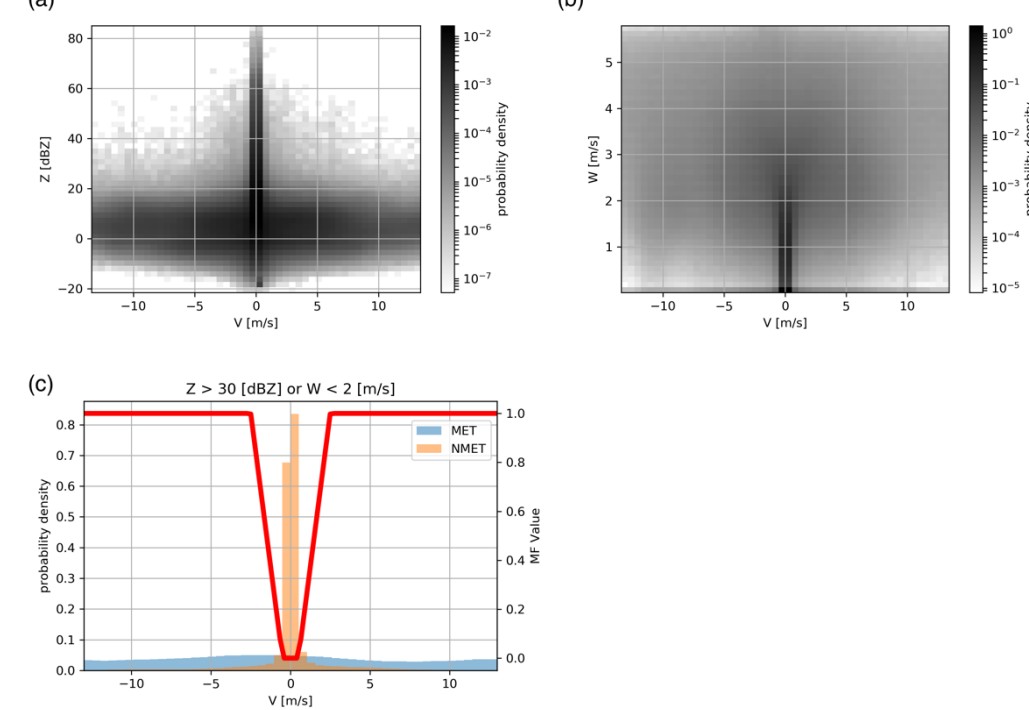

**Figure 5: (a) The 2D histogram of V vs. Z of NMET. (b) The 2D histogram of V vs. W of NMET. (c) The normalized frequency distribution and membership function of V after setting thresholds of Z and W.**


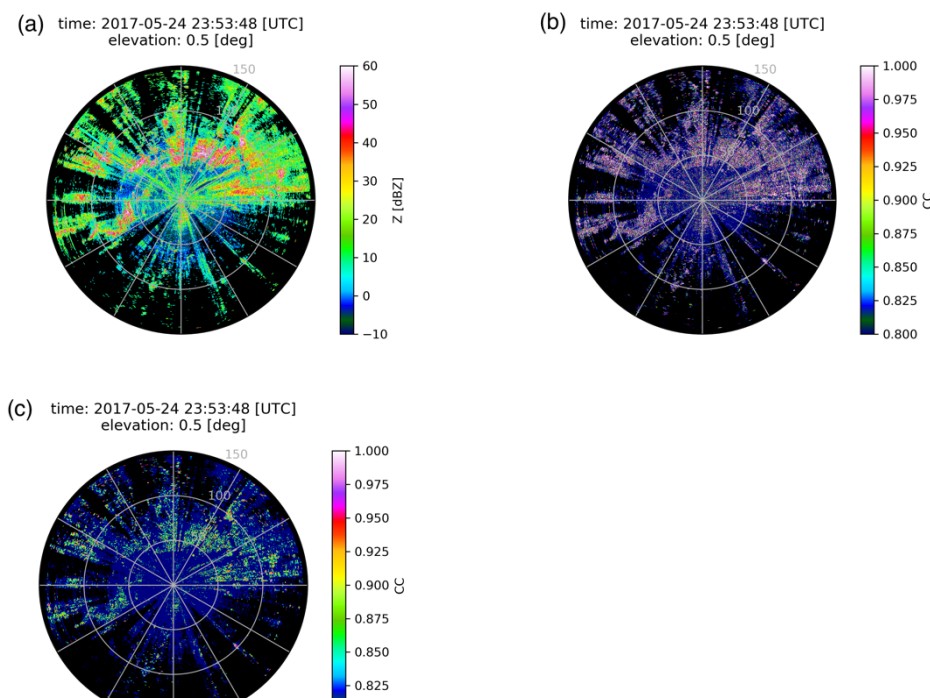


**Figure 6: (a) Z, (b) CC, and (c) CC after averaging along the radial using a 1.5-km window. All from the NUIST-CDP at 23:53 UTC 24 May 2017 from an elevation of 0.5°.**

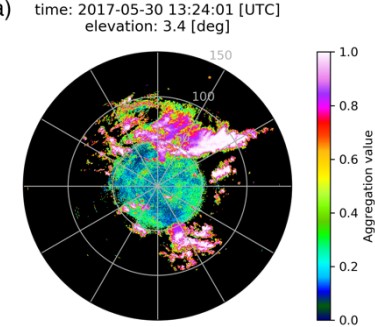

(a) time: 2017-05-30 13:24:01 [UTC]
elevation: 3.4 [deg]

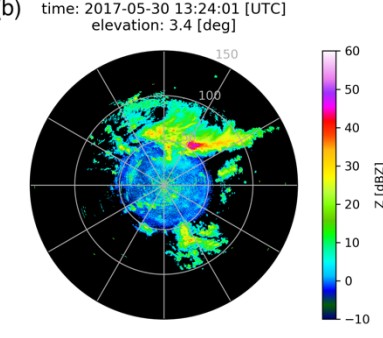

(b) time: 2017-05-30 13:24:01 [UTC]
elevation: 3.4 [deg]

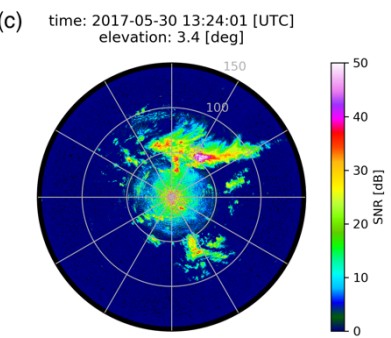

(c) time: 2017-05-30 13:24:01 [UTC]
elevation: 3.4 [deg]

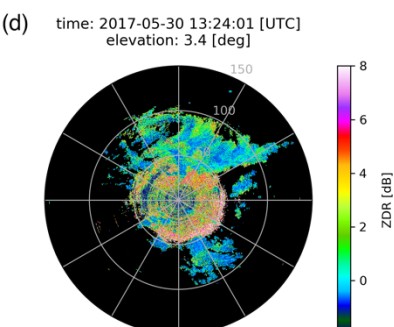

(d) time: 2017-05-30 13:24:01 [UTC]
elevation: 3.4 [deg]

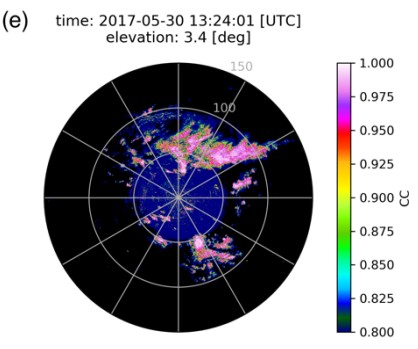

(e) time: 2017-05-30 13:24:01 [UTC]
elevation: 3.4 [deg]

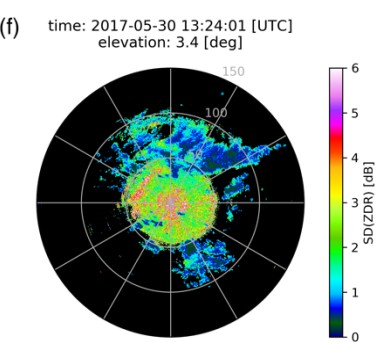

(f) time: 2017-05-30 13:24:01 [UTC]
elevation: 3.4 [deg]

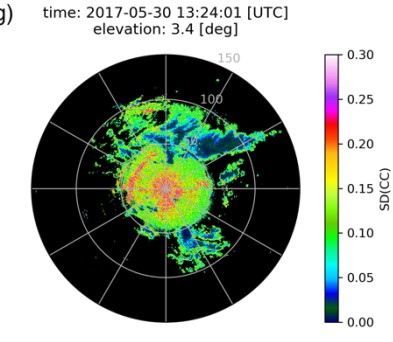

(g) time: 2017-05-30 13:24:01 [UTC]
elevation: 3.4 [deg]


**Figure 7: (a) A_MET (MetSignal), (b) Z, (c) SNR, (d) ZDR, (e) CC, (f) SD(ZDR), and (g) SD(CC). All from the NUIST-CDP at 13:24 UTC 30 May 2017 from an elevation of 3.4°.**

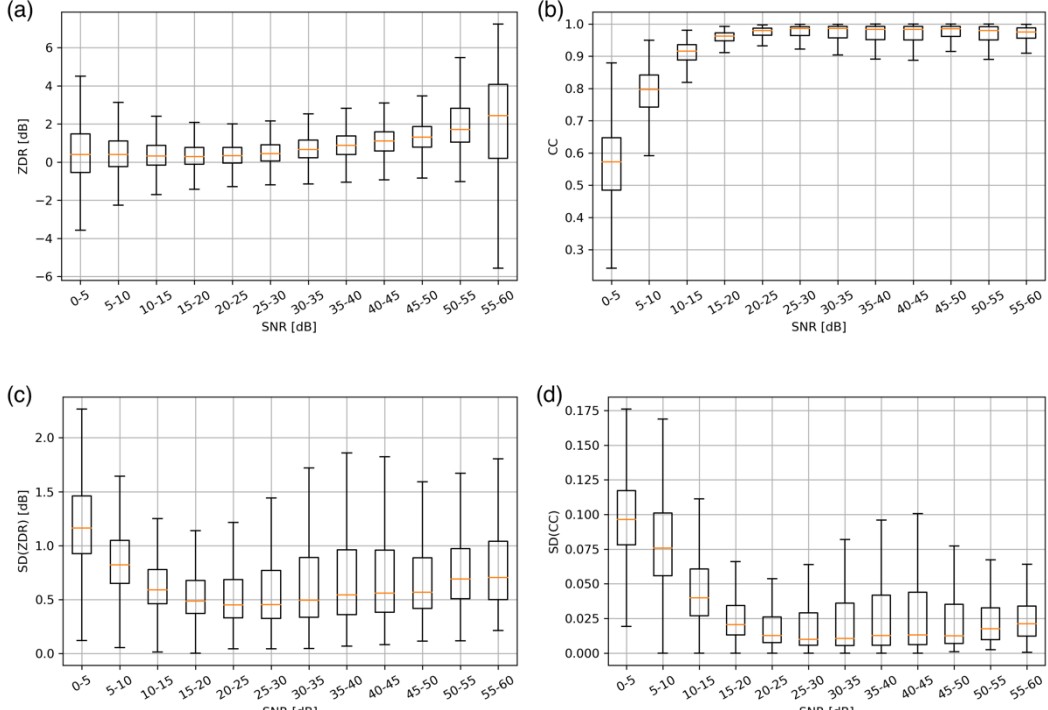

**Figure 8: The boxplot of SNR vs. polarimetric features of MET. (a) ZDR, (b) CC, (c) SD(ZDR), and (d) SD(CC).**


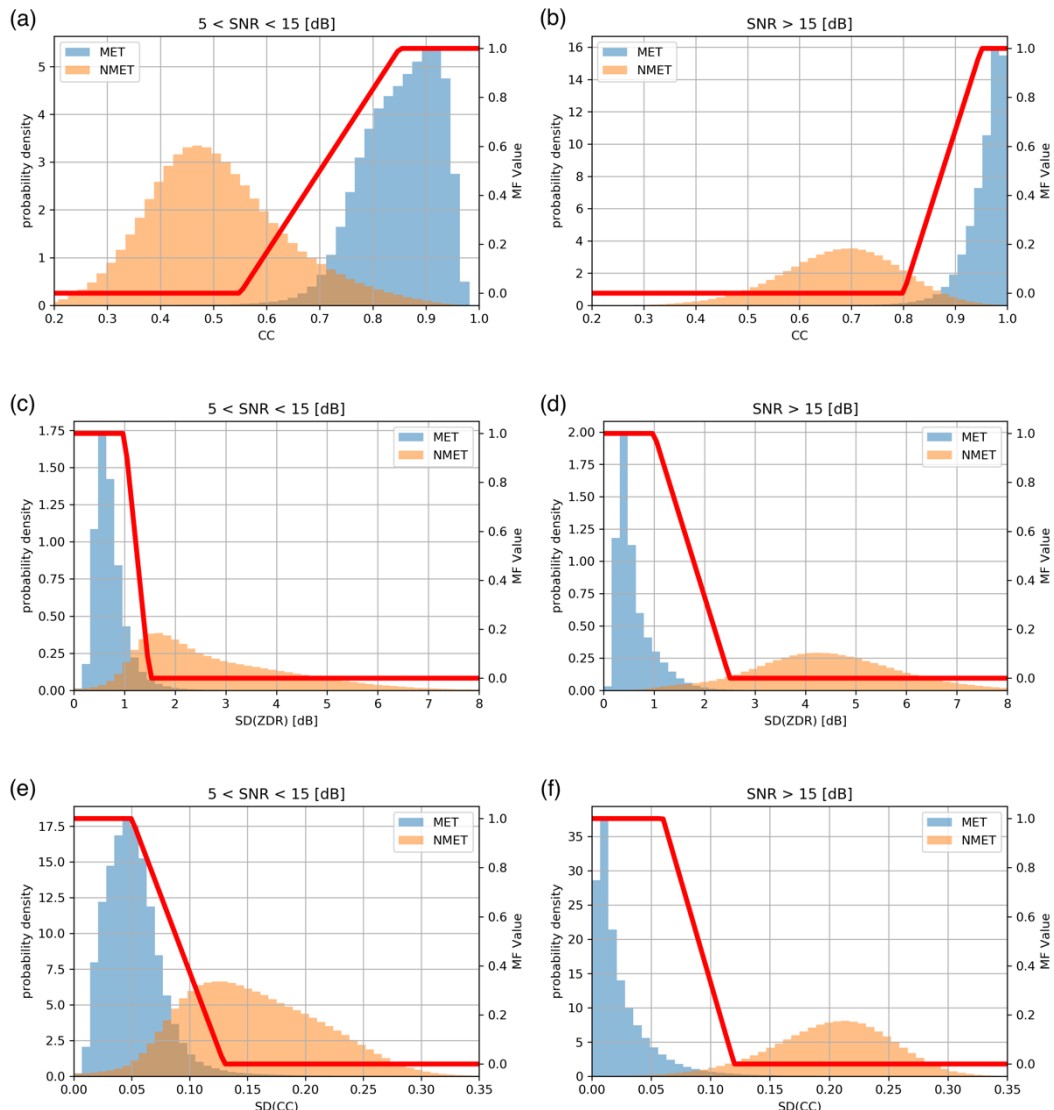

**Figure 9: The normalized frequency distributions and membership functions of polarimetric features stratified by SNR. (a) CC (5 < SNR < 15 dB), (b) CC (SNR >15 dB), (c) SD(ZDR) (5 < SNR < 15 dB), (d) SD(ZDR) (SNR >15 dB), (e) SD(CC) (5 < SNR < 15 dB), and (f) SD(CC) (SNR >15 dB).**



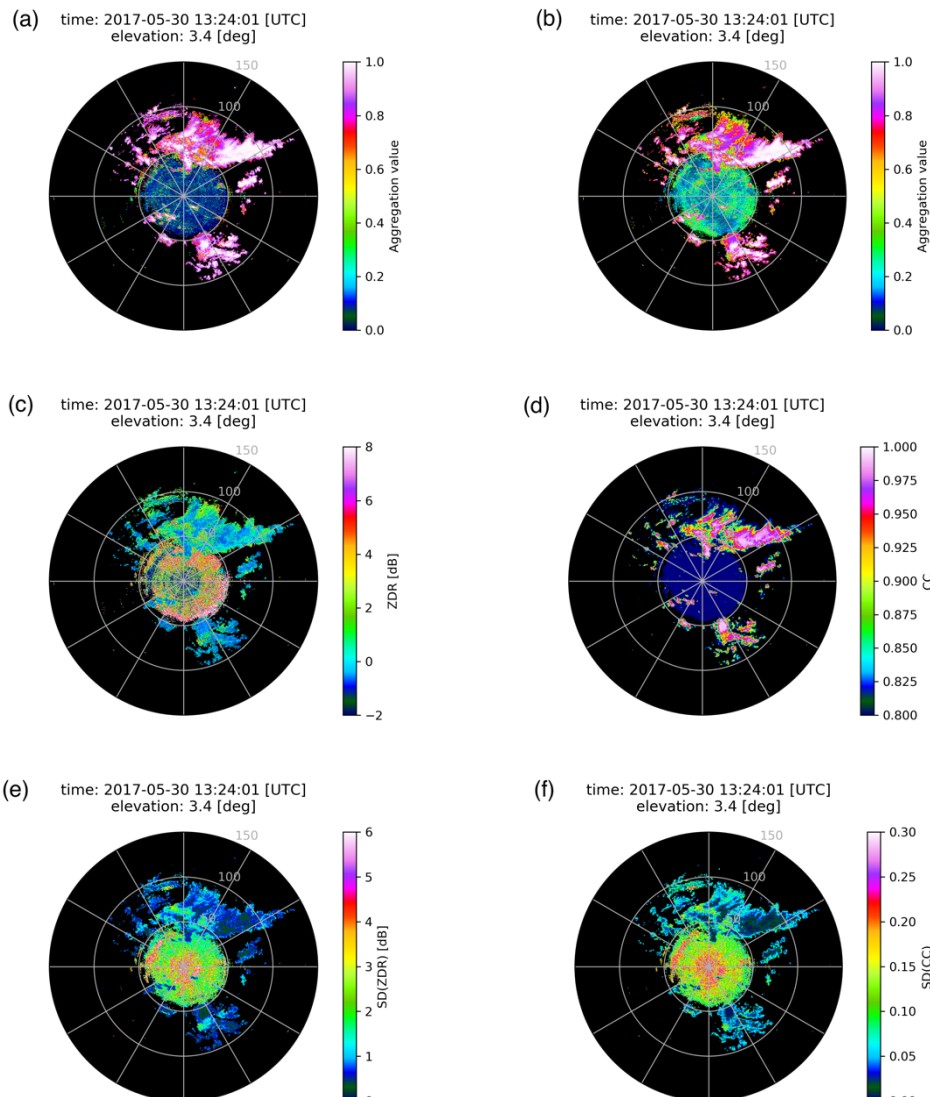

**Figure 10: (a) A$_{MET}$ (MetSignal_noise), (b) A$_{MET}$ (MetSignal) after masking SNR less than 5 dB, (c) ZDR after masking SNR less than 5 dB, (d) CC after averaging along the radial using a 1.5-km window and masking SNR less than 5 dB, (e) SD(ZDR) after masking SNR less than 5 dB, and (f) SD(CC) after masking SNR less than 5 dB. All from the NUIST-CDP at 1324 UTC 30 May 2017 from an elevation of 3.4°.**



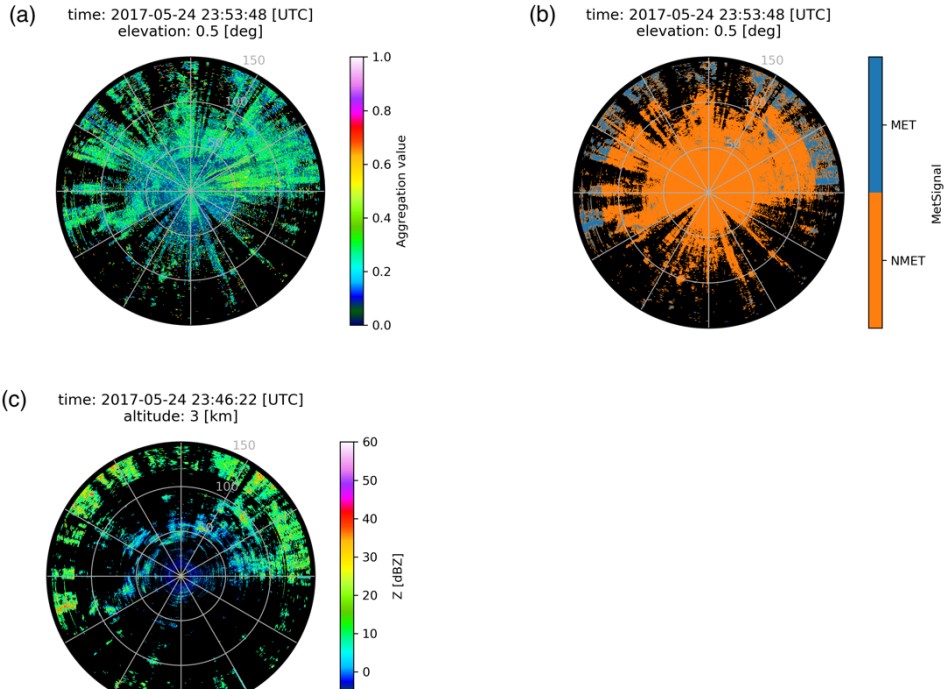

**Figure 11: (a) A<sub>MET</sub> (MetSignal), (b) MetSignal, and (c) The CAPPI of Z at the altitude of 3 km. (a) and (b) from the NUIST-CDP at 23:53 UTC 24 May 2017 from an elevation of 0.5° while (c) at 23:46 UTC 24 May 2017 (the previous volume scan of (a) and (b)).**


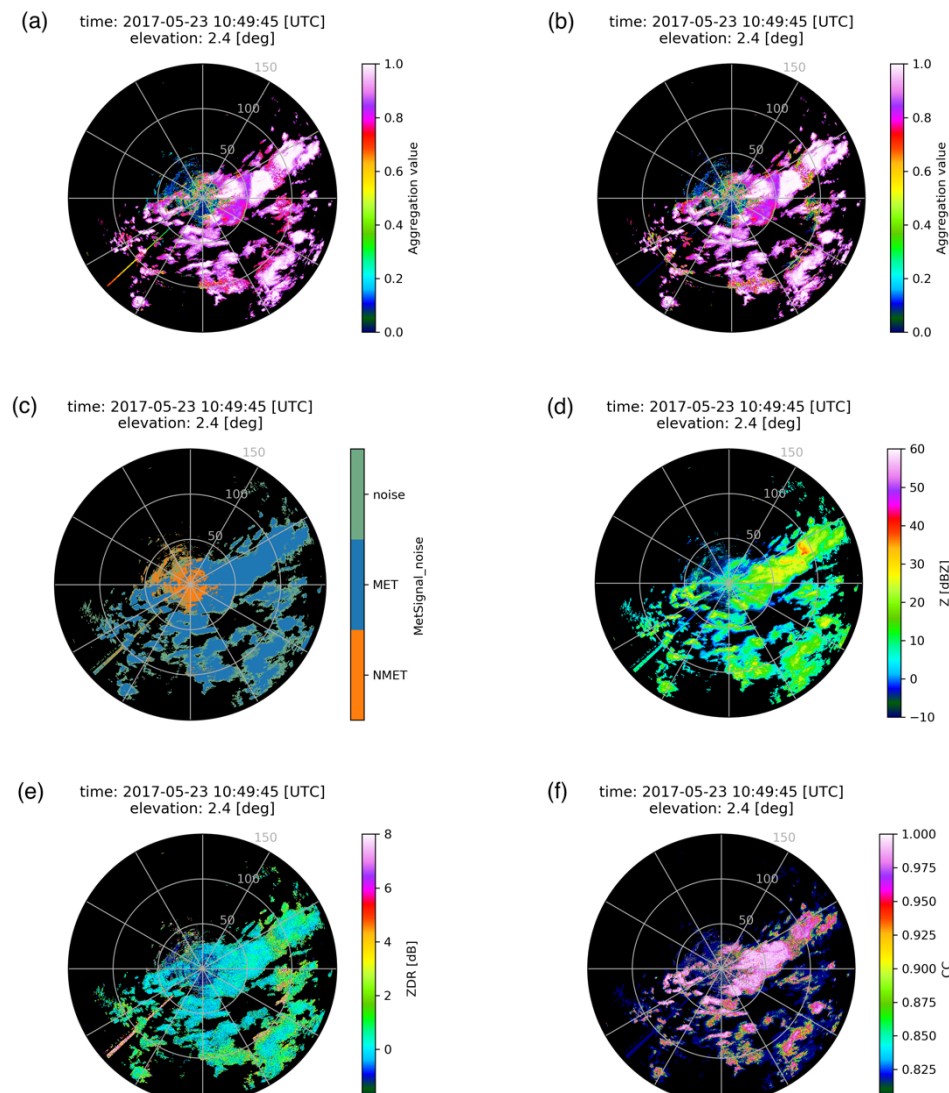

**Figure 12: (a) A<sub>MET</sub> (MetSignal_noise) after post-processing, (b) A<sub>MET</sub> (MetSignal_noise) before post-processing, (c) MetSignal_noise, (d) Z, (e) ZDR, and (f) CC. All from the NUIST-CDP at 10:49 UTC 23 May 2017 from an elevation of 2.4°.**