# Peer review of "Improved fuzzy logic method to distinguish between meteorological and non-meteorological echoes using C-band polarimetric radar data"

_Atmospheric Measurement Techniques, 2019_

## Referee Comment (RC1) · Anonymous Referee #1 · 23 Oct 2019

This paper proposes an improved version of the MeteSignal algorithm, introduced by Krause (2016), which separates meteorological (MET) and non-meteorological (NMET) echoes through dual-polarization radar features. The manuscript is structured into four sections: after a description of the MetSignal technique, the authors faced some impairments, related to the use of Doppler velocity, to the decrease of correlation coefficient in the region of ground clutter and anomalous propagation, to the accuracy of polarimetric variables in the weak signal region and to the melting layer region. For each of such issues, they proposed some improvements and adjustments that converge in a new version of the algorithm, labeled as MetSignal_noise. The latter, according to the results presented in Tab. 5, proved to the more efficient in the classification of MET

and NMET signals.

As a general comment, the paper can be considered of interest for operational radar meteorology field and seems adequate to the audience of Atmospheric Measurement Techniques journal, although it lacks in originality (nothing of really new is proposed). The methods used to improve MetSignal algortihm are decent and, in general, efficiently described.

However, the paper has been written with a style barely adequate for a scientific International Journal. Therefore, my first suggestion is to perform a formal revision of the entire manuscript and to pay more attention to the punctuation and to the English grammar. In this respect, the sentences at Lines 53-58, 92-94 and 191-194 need to be rephrased more clearly. Moreover, I suggest to reduce the number of Figures: as an example, Fig. 10 can be aggregated to Fig. 7 or Fig. 2.

Other issues, described in the following specific comments, need to be addressed by the authors.

- Method (Lines 97-99). It is not clear how the authors determined the preset thresholds (0.8 for warm season and 0.7 for cold season). Please provide a clarification and add some details in the revised manuscript.

- Method (Lines 112-117). The authors stated that a set of post-processing rules have been applied to adjust the classification results. I suggest to perform a sensitivity analysis to demonstrate the impact of such rules on the classification accuracy. What happens to the results of Tab. 5 if you do not apply the post-processing?

- Method (Line 198): I have some doubts about the criterion used to identify the melting layer. The latter, in this work, is defined as the region above 2.5 km. However, melting layer altitude can vary significantly from one meteorological event to another, because it is related to the freezing level height. Therefore, in my opinion, the authors must adopt a more objective criterion to detect melting layer in the training events. In this

respect, a careful analysis of horizontal reflectivity vertical profiles may be very useful. As the light of such remark, an update of Fig. 10 and, therefore, of Fig. 11, is needed.

- Method and Evaluation: I think that the authors should quantitatively assess the impact of the single impairments faced out in the Method section. In other words, I suggest to expand Tab. 5 and to present the results according to different scenarios: for example, what happens if you do not take into account the adjustment for Melting Layer Region in your algorithm?

- Evaluation. The authors must provide some clarifications about the contents of Table 5. How did you determine the classification accuracy percentage of MetSignal and MetSignal_noise algorithms? I assume you used the contingency table approach. This is an important point to clarify. Moreover, the performance of the two methods shall be evaluated with respect to more than one statistical score.

---

## Referee Comment (RC2) · Anonymous Referee #2 · 11 Nov 2019

This paper present an improvement of the MetSignal algorithm, an algorithm based on polarimetric C-band weater radar data to identify meteorological and nonmeteorological echoes.

A refinement of membership functions and the use of ad-hoc post-processing procedures increase the classification. Nevertheless some misclassification are still present and additional research is needed.

As general comments the paper is easy to read and figures are, normally, clear. The main weakness of the paper is section 4 "Evaluation". While it is acceptable the that the authors classify, by them self, events to develop and test the algorithm improvements (as done in section 3 based on table 3) the same is not so straight forward for the evaluation step. We assume that in the training MET and NMET echoes are well separated, so it is relatively simple verify that the algorithm outputs meet the expected "expert" classification. For the evaluation phase we hope that the authors use the improved algorithm in a more complex and mixed situation (like as probably happen in most events where MET and NMET are both present). If this is the case expert classification could be not so easy to do on a pixel basis. Further there is no evidence at all how is the "true" in these events compared and what the outputs of both algorithms. The evaluation results are present in table 5 where performances of both algorithms are shown. Here is not clear if the percentage are respect the "expert true" or what?

So this section need to be reformulated and a more deep analysis is needed.

Specific comments.

1) Section 3 line 104. Please indicate how events in table 4 are been classified.

2) Section 3.2.4 line 187 Please discuss how sensible is your method respect to time-schedule used. Since you use the "previous volume scan" at the "same location" to prevent misclassification of ML what happen if you have a quite fast-moving system?

3) section 4 line 215 Please explain how are classified the events in table 4.

4) Section "conclusion" line 232 Correct the section number

5) references linea 265-266 Please complete the reference

6) references lines 282-283 Please add the journal

7) figure 5 In Panel (a) what means "aggregation value"? Please explain the color scale ?

8 ) figure 8 As for point 7 related to panles (a) and (b)

9) figure 11 As above Further, in order to increase the readability of this figure could

you evaluate to display a zoomed area around the ML at 100 km range from radar in the SE quadrant?

---

## Author Comment (AC1) · 22 Nov 2019

Authors are grateful to the reviewer for a careful reading of the work and valuable comments. The responses are in the attached file (Supplement).

Please also note the supplement to this comment: https://www.atmos-meas-tech-discuss.net/amt-2019-337/amt-2019-337-AC1-supplement.zip
* * *

---

## Author Comment (AC2) · 22 Nov 2019

Authors are grateful to the reviewer for a careful reading of the work and valuable comments. The responses are in the attached file.

Please also note the supplement to this comment: https://www.atmos-meas-tech-discuss.net/amt-2019-337/amt-2019-337-AC2-supplement.zip
* * *

---

## Author Response (AR1)

**Responses to Anonymous Referee #1**

**Comments 1**

However, the paper has been written with a style barely adequate for a scientific International Journal. Therefore, my first suggestion is to perform a formal revision of the entire manuscript and to pay more attention to the punctuation and to the English grammar. In this respect, the sentences at Lines 53-58, 92-94 and 191-194 need to be rephrased more clearly.

**Response**

First of all, I would like to apologize for my poor English writing. A complete and careful revision has been completed as recommended by the reviewer (corresponding to Lines 8-16, 68-69, 75-77, 82-84, 136-139, 145-147, 175-177, and 220-223 in the revised manuscript respectively), and the special attention was paid to sentences at Lines 53-58, 92-94, and 191-194 (corresponding to Lines 53-59, 92-95, and 209-212 in the revised manuscript respectively).

**Comments 2**

Moreover, I suggest to reduce the number of Figures: as an example, Fig. 10 can be aggregated to Fig. 7 or Fig. 2.

**Response**

Reviewer's comment is adopted and Fig. 10 is incorporated into Fig. 2 (it's Fig. 2f in revised manuscript).

Method (Lines 97-99). It is not clear how the authors determined the preset thresholds (0.8 for warm season and 0.7 for cold season). Please provide a clarification and add some details in the revised manuscript.

**Response**

This comment is aimed at Section 3.1, whose role in the paper is to describe the overall framework of MetSignal algorithm. Therefore, the thresholds (i.e. 0.8 for warm season and 0.7 for cold season) used in MetSignal\_noise algorithm before are those adopted in Krause (2016). However, the reviewer's comment reminds me that this threshold should be a local value (similar to the membership functions) and needs statistical analysis to get the optimal result. Therefore, an objective statistical method is adopted to determine this threshold and details have been added in the revised manuscript (corresponding to Lines 113-118 and Lines 198-202). The statistics results in Table 5 have also been updated due to changes in thresholds. Figure 3 (index in the revised manuscript) is added.

**Comments 4**

Method (Lines 112-117). The authors stated that a set of post-processing rules have been applied to adjust the classification results. I suggest to perform a sensitivity analysis to demonstrate the impact of such rules on the classification accuracy. What happens to the results of Tab. 5 if you do not apply the post-processing?

**Response**

A sensitivity analysis has been performed under the suggestion by reviewer, and the results are shown in the following table. Two methods were used to compute skill: the fraction of correctly classification for each echo type (FCC) and the overall Heidke skill score (HSS) (Lines 234–239 in revised manuscript for details). By comparing the classification performance of the MetSignal algorithm before and after post-processing, it can be found that the classification skill of MetSignal algorithm in MET (NMET) will decrease (increase) without post-processing, which is due to the lack of special precaution in the ML regions and reduction of misclassification between AP and MET caused by post-processing respectively.

|                         | FCC MET | FCC NMET | HSS  |
|-------------------------|--------------------|---------------------|------|
| with post-processing    | 86.8 %             | 96.2 %              | 0.83 |
| without post-processing | 84.8 %             | 96.7 %              | 0.81 |

Although the results of the above sensitivity analysis prove the necessity of postprocessing (HSS increases from 0.81 to 0.83), I think this sensitivity analysis is not necessary in this paper. Just like the response to comment 3, the role of Section 3.1 in this paper is to describe the overall framework of MetSignal algorithm, and the postprocessing rules mentioned in Section 3.1 were not proposed by authors but an integral part of the raw MetSignal algorithm (Krause 2016). However, in order to achieve the same purpose of the sensitivity analysis suggested by the reviewer (i.e., to prove the importance of post-processing rules), a typical case has been added in the revised manuscript (corresponding to Lines 124-129). Figure 4 (index in the revised manuscript) is added.

Absolutely, if the reviewer still insists on adding this part to the paper, I'll make some adjustments.

**Comments 5**

Method (Line 198): I have some doubts about the criterion used to identify the melting layer. The latter, in this work, is defined as the region above 2.5 km.

However, melting layer altitude can vary significantly from one meteorological event to another, because it is related to the freezing level height. Therefore, in my opinion, the authors must adopt a more objective criterion to detect melting layer in the training events. In this respect, a careful analysis of horizontal reflectivity vertical profiles may be very useful. As the light of such remark, an update of Fig. 10 and, therefore, of Fig. 11, is needed.

**Response**

The doubt proposed by reviewer about the altitude of melting layer is very insightful. The melting layer altitude does vary significantly from one meteorological event to another, and the lack of accurate and objective positioning of melting layer altitude is one of the defects of MetSignal\_noise (as mentioned in Section 5). Therefore, at present, only through the analysis of existing data to determine the approximate location of the melting layer altitude, and need to adjust with the season and location. The altitude threshold of potential melting layer defined in this paper (2.5 km) is the result after analyzing the training set (i.e., all regions affected by the melting layer are higher than 2.5 km in the training set), and details have been added in the revised manuscript (corresponding to Lines 216-217). Therefore, the statistical result in Fig. 10 should be credible.

As can be seen from the title of Krause (2016), the core idea of MetSignal algorithm is as simple as possible on the premise of effectiveness. Therefore, it is obviously not suitable to use a melting layer identification method with the complexity comparable to the whole MetSignal algorithm. At present, I am trying to find a simple method to automatically obtain the melting layer altitude. The idea proposed by the reviewer that positioning the melting layer altitude by the horizontal reflectivity vertical profiles is of great reference value.

Method and Evaluation: I think that the authors should quantitatively assess the impact of the single impairments faced out in the Method section. In other words, I suggest to expand Tab. 5 and to present the results according to different scenarios: for example, what happens if you do not take into account the adjustment for Melting Layer Region in your algorithm?

**Response**

Reviewer's comment is adopted. The sensitivity analysis for the four improvement steps mentioned in Section 3.2 has been added in the revised manuscript (corresponding to Lines 256-274), and Table 5 is also expanded as suggested by reviewers.

**Comments 7**

Evaluation. The authors must provide some clarifications about the contents of Table 5. How did you determine the classification accuracy percentage of MetSignal and MetSignal\_noise algorithms? I assume you used the contingency table approach. This is an important point to clarify. Moreover, the performance of the two methods shall be evaluated with respect to more than one statistical score.

**Response**

Reviewer's comment is adopted. An additional skill score (HSS) is added, and the score has been used in Table 5 is described in detail in the revised manuscript (corresponding to Lines 234-239).

**Responses to Anonymous Referee #2**

**Comments 1**

The main weakness of the paper is section 4 "Evaluation". While it is acceptable the that the authors classify, by them self, events to develop and test the algorithm improvements (as done in section 3 based on table 3) the same is not so straight forward for the evaluation step. We assume that in the training MET and NMET echoes are well separated, so it is relatively simple verify that the algorithm outputs meet the expected "expert" classification. For the evaluation phase, we hope that the authors use the improved algorithm in a more complex and mixed situation (like as probably happen in most events where MET and NMET are both present). If this is the case expert classification could be not so easy to do on a pixel basis. Further there is no evidence at all how is the "true" in these events compared and what the outputs of both algorithms. The evaluation results are present in table 5 where performances of both algorithms are shown. Here is not clear if the percentage are respect the "expert true" or what? So this section need to be reformulated and a more deep analysis is needed.

**Response**

The doubt proposed by reviewer about the credibility of training/test set is very insightful, which is a very difficult (unresolved) problem in the field of radar meteorology. The details of this problem are as follows. Generally speaking, the evaluation methods of echo classification are generally divided into two categories: qualitative and quantitative. The qualitative evaluation method generally adopts the way of showing typical cases (Krause 2016), whose disadvantage is that it cannot show the universality of the algorithm, and the cases shown may have the suspicion of luck. The quantitative evaluation method is generally evaluated by the performance of the algorithm on the selected test set in advance (i.e., the expert classification as the reviewer said; Lakshmanan et al. 2014,

Tang et al. 2014). In addition, it can also be evaluated by comparing the performance of meteorological application (e.g., quantitative precipitation estimation (QPE)) before and after echo classification (Cho et al. 2006), whose disadvantage is that other error sources will be introduced and also affect the performance of QPE, such as the instability of Z-R relationship and radar miscalibration. Therefore, this paper uses the method of expert classification for evaluation, which is widely used at present. There are two methods to select training set and test set. The first method is to select cases that only have single echo type in the whole sweep; once there are multiple echo types mixed in a case, it will be discarded (Grecu and Krajewski 2000). This method is relatively simple and convenient, and more suitable for sufficient data. The disadvantage of this method is that it cannot extract the cases with multiple echo types mixed together, which has a high frequency of occurrence as the reviewer said. The second method is to extract data through graphical user interface (GUI; Lakshmanan et al. 2007, Rennie et al. 2015), which is the method adopted in this paper. The GUI program is developed by the author, and its interface is shown in the figure below (left). Mark the region where the data is to be extracted with the mouse, and store the four vertices of the region in the CSV file (as shown in the figure below (right)). Although the software looks "rough", it's enough to complete my research.

At present, it is very difficult to verify the truth of the training/test set. This is because there are not enough observations as evidence, so it can only rely on subjective examine by manual. However, this is not only the problem that this paper faces, but also a problem that all the researches related to radar echo classification face, especially to the hydrometeor classification (Park et al. 2009, Zrnic et al. 2001). The classification between MET and NMET is relatively easier than that of the hydrometeor classification, because it has obvious criterions in some cases, such as data from satellite and rain gauge can prove when is clear air. As proposed by reviewers, it is necessary to evaluate the algorithm in a more complex and mixed situation (e.g., use the GUI program mentioned above to extract the core of the convective storm embedded in the clear-air echoes). However, it is very difficult and unrealistic to clearly distinguish between MET and NMET at the boundary between them by manual classification (other means is even more impossible), which will inevitably lead to subjective bias. Therefore, the author tries to avoid selecting too "complex" data (e.g., the boundary between MET and NMET) to ensure that no subjective bias is introduced, which is also the default view of all related researches (Lakshmanan et al. 2014, Tang et al. 2014). Therefore, the percentages shown in Table 5 are indeed "expert truth". But this "expert truth" is obtained from the evaluation results of the "not very complex" (without dispute and subjective bias) test set.

The author complements and improves the paper according to the reviewer's suggestion, especially in the Section 4, including how to select training/test set (same as the second and fourth comments; corresponding to Lines 102-106 and 235-236 in the revised manuscript), adding an overall skill score method (HSS; corresponding to Lines 234-239 in the revised manuscript), and analyzing the sensitivity of the improvement steps mentioned in Section 3.2 (corresponding to Lines 256-274 in the revised manuscript).

**Comments 2**

Section 3 line 104. Please indicate how events in table 4 are been classified.

**Response**

Reviewer's comment is adopted and how to select training set has been supplemented in the revised manuscript (corresponding to Lines 102-106).

Section 3.2.4 line 187 Please discuss how sensible is your method respect to timeschedule used. Since you use the "previous volume scan" at the "same location" to prevent misclassification of ML what happen if you have a quite fast-moving system?

**Response**

The part questioned by the reviewer (i.e., use the "previous volume scan" at the "same location" to prevent misclassification of ML) appears in Section 3.1 for the first time (corresponding to Lines 122-124). The role of Section 3.1 in this paper is to describe the overall framework of MetSignal algorithm, and the post-processing rule (use the "previous volume scan" at the "same location" to prevent misclassification of ML is one of the three post-processing rules) mentioned in Section 3.1 was not proposed by authors but an integral part of the raw MetSignal algorithm (Krause 2016). Just like the question raised by the reviewer, I'm also confused about it. Fortunately, however, this part has been removed in the improved algorithm (i.e., MetSignal\_noise) due to the potential risk of misclassifying AP into MET (corresponding to Lines 204–212 for details).

**Comments 4**

section 4 line 215 Please explain how are classified the events in table 4.

**Response**

Reviewer's comment is adopted and how to select test set has been supplemented in the revised manuscript (corresponding to Lines 235-236).

Section "conclusion" line 232 Correct the section number.

**Response**

Reviewer's comment is adopted, and the section number has been corrected in the revised manuscript (corresponding to Line 275).

**Comments 6**

references lines 265-266 Please complete the reference.

**Response**

Reviewer's comment is adopted, and the reference has been completed in the revised manuscript (corresponding to Lines 312-313).

**Comments 7**

references lines 282-283 Please add the journal.

**Response**

Reviewer's comment is adopted, and the missing journal name has been added in the revised manuscript (corresponding to Lines 329-330).

figure 5 In Panel (a) what means "aggregation value"? Please explain the color scale?

**Response**

The "aggregation value" questioned by the reviewer appears in Section 3.1 for the first time (corresponding to Lines 92-98), which is a terminology in fuzzy logic technology (Zrnic et al. 2001). The formula for the calculation of aggregate value for the MET ( $A_{MET}$ ) is shown in Eq.1 (the aggregation value for the NMET can be computed by subtracting  $A_{MET}$  from 1). The  $A_{MET}$  of a specific pixel indicates the possibility that this pixel is MET, in other words, 1 in the color scale represents 100% (very likely to be MET), 0 represents 0% (next to impossible to be MET). After getting the  $A_{MET}$  by the calculation of Eq.1, compare it with a preset threshold. The target pixel will be classified as MET if  $A_{MET}$  exceeds the threshold; otherwise, it will be classified as NMET (corresponding to Lines 113–118 for details).

**Comments 9**

figure 8 As for point 7 related to panels (a) and (b).

**Response**

As for response to comment 8.

figure 11 As above Further, in order to increase the readability of this figure could you evaluate to display a zoomed area around the ML at 100 km range from radar in the SE quadrant?

**Response**

As for response to comment 8.

As shown in the figure below, Fig. 11 has been modified according to the reviewer's comments. However, revised Fig. 11 may raise some readers' doubts about what happened in other quadrants (whether the authors want to hide something in the rest of this case). Therefore, the author suggests that the reviewer reconsider whether to make this modification. Absolutely, if the reviewer still insists on making this modification, I'll make some adjustments.

---

## Referee Report (RR1)

This paper present an improvement of the MetSignal algorithm, an algorithm based on polarimetric C-band weater radar data to identify meteorological and nonmeteorological echoes.

A refinement of membership functions and the use of ad-hoc post-processing procedures increase the classification. Nevertheless some misclassification are still present and additional research is needed.

As general comments the paper is easy to read and figures are, normally, clear.
Most of the weak points was addressed during the open discussion so only few minor comments are present now.

1) ZDR filter – Page 4 line 121.
In C band resonance effects which increase substantially ZDR are present so it is possible that the filter reject not only CA but also weather echoes.
Such ZDR values could be reached if wet hailstone is inside the radar volume and in the so-called ZDR column.

2) CC range averaging – Page 5 Lines 149-155
What's happen in boundary region between MET e NMET ?

---

## Author Response (AR2)

1. in view of the contribution of Zhigang Chu and Xiaoran Zhuang in the revision, they were added as co-authors (Line 4 in the revised manuscript)
2. adjustments of the authors' affiliations (Lines 5-7 in the revised manuscript)
3. details about the distance averaging of CC in the boundary region between MET and NMET (Lines 160-163 in the revised manuscript)
4. details about how to mitigate the problem caused by wet hailstone (Lines 238-241 in the revised manuscript)
5. details about the future development of more reliable method to estimate ML Lines 82-84 are rephrased (Lines 300-301 and 304-308 in the revised manuscript)
6. add the acknowledgements (Lines 312-315 in the revised manuscript)
7. add the references (Lines 339-342 and 384-385 in the revised manuscript)